# Genomic Epidemiology of SARS-CoV-2 in Tocantins State and the Diffusion of P.1.7 and AY.99.2 Lineages in Brazil

**DOI:** 10.3390/v14040659

**Published:** 2022-03-23

**Authors:** Ueric José Borges de Souza, Raíssa Nunes dos Santos, Fernando Lucas de Melo, Aline Belmok, Jucimária Dantas Galvão, Tereza Cristina Vieira de Rezende, Franciano Dias Pereira Cardoso, Rogério Fernandes Carvalho, Monike da Silva Oliveira, Jose Carlos Ribeiro Junior, Evgeni Evgeniev Gabev, Ester Cerdeira Sabino, Clarice Weis Arns, Bergmann Morais Ribeiro, Fernando Rosado Spilki, Fabrício Souza Campos

**Affiliations:** 1Bioinformatics and Biotechnology Laboratory, Campus of Gurupi, Federal University of Tocantins, Gurupi 77410-570, Brazil; engraissanunes@gmail.com; 2Baculovirus Laboratory, Department of Cell Biology, Institute of Biological Sciences, University of Brasilia, Brasília 70910-900, Brazil; flucasmelo@gmail.com (F.L.d.M.); abelmokadias@gmail.com (A.B.); bergmann.ribeiro@gmail.com (B.M.R.); 3Central Public Health Laboratory of the State of Tocantins, Palmas 77054-970, Brazil; jucydg@mail.uft.edu.br (J.D.G.); tereza.rezende@gmail.com (T.C.V.d.R.); francianocardoso@yahoo.com.br (F.D.P.C.); 4Laboratory of Microbiology, Federal University of North Tocantins, Araguaína 77804-970, Brazil; fcrogeriofc@hotmail.com (R.F.C.); monikeoliveira@discente.ufg.br (M.d.S.O.); ribeirojuniorjc@gmail.com (J.C.R.J.); 5Molecular Biology Laboratory, Institute of Biological Sciences, Federal University of Goiás, Goiânia 74001-970, Brazil; 6Department of Physiology and Pathophysiology, Medical University of Sofia, 1431 Sofia, Bulgaria; egabev@medfac.mu-sofia.bg; 7Department of Infectious and Parasitic Diseases, Institute of Tropical Medicine, Faculty of Medicine, University of São Paulo, São Paulo 05403-000, Brazil; sabinoec@gmail.com; 8Laboratory of Animal Virology, Institute of Biology, University of Campinas, Campinas 13083-862, Brazil; clarns@gmail.com; 9Molecular Microbiology Laboratory, Feevale University, Novo Hamburgo 93525-075, Brazil; fernandors@feevale.br

**Keywords:** SARS-CoV-2, variants of concern, genome analysis, viral evolution, viral trajectory

## Abstract

Tocantins is a state in the cross-section between the Central-West, North and Northeast regions of Brazilian territory; it is a gathering point for travelers and transportation from the whole country. In this study, 9493 genome sequences, including 241 local SARS-CoV-2 samples (collected from 21 December 2020, to 16 December 2021, and sequenced in the MinION platform) were analyzed with the following aims: (i) identify the relative prevalence of SARS-CoV-2 lineages in the state of Tocantins; (ii) analyze them phylogenetically against global SARS-CoV-2 sequences; and (iii) hypothesize the viral dispersal routes of the two most abundant lineages found in our study using phylogenetic and phylogeographic approaches. The performed analysis demonstrated that the majority of the strains sequenced during the period belong to the Gamma P.1.7 (32.4%) lineage, followed by Delta AY.99.2 (27.8%), with the first detection of VOC Omicron. As expected, there was mainly a dispersion of P.1.7 from the state of São Paulo to Tocantins, with evidence of secondary spreads from Tocantins to Goiás, Mato Grosso, Amapá, and Pará. Rio de Janeiro was found to be the source of AY.99.2 and from then, multiple cluster transmission was observed across Brazilian states, especially São Paulo, Paraiba, Federal District, and Tocantins. These data show the importance of trade routes as pathways for the transportation of the virus from Southeast to Northern Brazil.

## 1. Introduction

The ongoing COVID-19 pandemic is one of the greatest global threats in modern history. Since 2019, with the host spillover of the Severe Acute Respiratory Syndrome Coronavirus 2 (SARS-CoV-2) [1], the novel betacoronavirus has had a severe impact on human health, the economy, and social life. This global scenario demonstrated the limitations of One Health policies in terms of minimizing transmission across continents. The lack of measures to control the new virus in Brazil included inadequate public interventions, non-scientific communications, social inequality, and slow vaccination [2,3]. Moreover, global transit networks increase the risk of importing and exporting pathogens, as was evidenced by the circulation of variants of concern (VOCs) around the world during the pandemic [4,5].

International public efforts have been focused on the emergence of virus variants leading to changes in viral fitness. The mutations present in SARS-CoV-2 variants may increase transmissibility, enhance escape from the human immune response, or alter biologically important phenotypes in a way that confers a fitness advantage, such as the mutations of the spike affecting antigenicity [6]. The global emergence of several VOCs, recently renamed by the World Health Organization (WHO), including the UK variant (Alpha or B.1.1.7), the South Africa variant (Beta or B.1.351), the Brazil variant (Gamma or P.1), and the India variant (Delta or B.1.617.2), highlights the utmost importance of genomic surveillance for monitoring viral mutations [7,8]. Gamma and Delta evolved into new sublineages, called P1.X and AY.X, respectively [9,10].

Brazil faced one of the highest numbers of confirmed SARS-CoV-2 cases worldwide, 27,968,811 with an incidence of 14,005.5 cases per 100,000 inhabitants (Brazilian Ministry of Health, https://covid.saude.gov.br, accessed on 15 March 2022). Tocantins is a state located in a very peculiar region of Brazil; it is a gathering point for transportation routes from the Southeast, Central-Western, North, and Northeast regions of the country (Figure 1). Tocantins presents two of the most impacted Brazilian biomes, namely Cerrado (88% of the state area) and Amazon (12%), and has been highlighted as the most recent deforestation hotspot related to soy and livestock production [11]. Soybean cultivation is spread throughout Tocantins, which favors the movement of people. In addition, the proximity of Tocantins state to the Amazonas state facilitated the spread of the Gamma variant [3]. The first patient infected with SARS-CoV-2 in Tocantins was reported on 16 March 2020. Since then, the state has reported 300,957 cases and 4132 deaths, with an incidence of 19,134.30 cases per 100,000 inhabitants (Integra Saúde Tocantins, http://integra.saude.to.gov.br/covid19/InformacoesEpidemiologicas, accessed on 15 March 2022).

Genomic surveillance provides important clues as to virus–host dynamics [12,13]. Long-read sequencing was first used in arboviral genomic studies [14,15] and then quickly adapted for SARS-CoV-2 given the public health emergency. Thus, next-generation sequencing (NGS) technologies represent a powerful tool for tracing the origin, spread, and transmission chains of outbreaks, as well as monitoring the evolution of etiological agents. The COVID-19 pandemic has triggered efforts for real-time surveillance strategies based on viral genome sequencing. In Brazil, the Corona-ômica-BR project (a network made up of 11 research institutions and dozens of researchers) is harnessing efforts and human resources to track the viral spread and support public health authorities. In this study, we report the complete genome sequences and phylogenetic analysis of 241 SARS-CoV-2 genomes detected in the Tocantins state, located in the North region of Brazil, from December 2020 to December 2021. Furthermore, the first description of SARS-CoV-2 molecular epidemiology in this Brazilian state is presented.

## 2. Materials and Methods

### 2.1. Sample Collection

Samples from patients with SARS-CoV-2-positive nasopharyngeal RT-qPCR were collected between 21 December 2020, and 16 December 2021, at the Central Public Health Laboratory of the State of Tocantins (Figure 1). Reverse-transcription-quantitative real-time polymerase chain reaction (RT-qPCR) for SARS-CoV-2 is the gold-standard method for laboratory diagnosis of COVID-19. The kit used for all samples was the Allplex 2019-nCoV assay, Seegene Inc., Seoul, Korea (provided by the Brazilian Ministry of Health and used as a routine procedure at all public health laboratories in Brazil). The RT-qPCR reactions were conducted using 6.5 μL of RNA input, instead of the recommended 8 μL. Briefly, the Allplex kit employs target genes and labeled probes E (FAM), N (Quasar 670), RdRP (Cal Red 610), and an internal control (IC; HEX) in one multiplex RT-qPCR. All RT-qPCR reactions were performed using the QuantStudio 5 Real-Time PCR System (Applied Biosystems, Waltham, CA, USA). In total, 241 positive samples with quantification cycle (Cq) below 25 for at least one primer were selected and submitted to the SARS-CoV-2 genome sequencing. The study was approved by the Ethics Committee (33202820.7.1001.5348).

### 2.2. RNA Extraction and Sequencing

The samples were processed at the Central Public Health Laboratory of the State of Tocantins, with an Extracta kit Viral RNA MVXA-P096 FAST (Loccus, Brazil) using an automated extractor (Extracta 96, Loccus, Brazil) following the manufacturer’s guidelines. The cDNA synthesis was performed using Luna Script RT SuperMix (5×) (New England Biolabs, Ipswich, MA, USA). The synthesized cDNAs were used as templates for a multiplexed PCR reaction using the two non-overlapping primer pools provided by the ARTIC Network to generate ~400 bp amplicons tiled across the genome (V3 nCov-2019 primers) (ARTIC primer set). Amplicons from both primer pools were combined and purified with a 1× volume of Ampure XP beads (Beckman Coulter, Brea, CA, USA). The MinION library preparation was performed using the Ligation Sequencing kit SQK-LSK-109 and Natives Barcoding kits EXP-NBD104 and EXP-NBD114 (Oxford Nanopore, Oxford, UK). The resulting library was loaded on R9.4 Oxford MinION flowcells (FLO-MIN106) and sequenced using the MinION Mk1B device. ONT MinKNOW software was used to collect raw data. High-accuracy base-calling and quality control analyses were performed using Guppy (v6.0.1) and NanoPlot (v.1.33.0), respectively. Assembly of the high accuracy base called fastq files was performed using the nCoV-2019 novel coronavirus bioinformatics protocol (https://artic.network/ncov-2019/ncov2019-bioinformatics-sop.html, accessed on 12 January 2022) with Minimap2 [16] and Medaka (https://github.com/nanoporetech/medaka, accessed on 12 January 2022) for consensus sequence generation.

### 2.3. Comparative Genome Analysis

All sequences were analyzed with Nextclade v1.7.2 [17] (https://clades.nextstrain.org/, accessed on 12 January 2022) to determine the clade and the number of gap regions. Lineages were assigned to each genome using Pangolin v3.1.17 [18] (https://github.com/cov-lineages/pangolin, accessed on 12 January 2022). The sequences were grouped according to the two most common VOCs (Gamma and Delta) and aligned using MAFFT v7.490 [19], with the strain Wuhan-Hu-1 genome reference (accession number: NC_045512.2) as the first sequence in the alignment. Genome map and SNP histogram were generated for each grouped lineage using the msastats.py script, and plotAlignment and plotSNPHist functions [20] (https://github.com/laduplessis/SARS-CoV-2_Guangdong_genomic_epidemiology/, accessed on 12 January 2022).

In order to identify single-nucleotide polymorphisms (SNPs) and insertions/deletions (INDELs), the assembled genomes were mapped against the reference (NC_045512.2) separately according to grouped lineages by Snippy variant calling and core genome alignment v4.6.0 (https://github.com/tseemann/snippy, accessed on 12 January 2022). This pipeline uses FreeBayes v1.3.2 [21] to call variants and snpEff v5.0 [22] to annotate variants based on their genomic locations and predicts coding effects such as synonymous or non-synonymous amino acid replacement.

### 2.4. Phylogenetic Analysis

The 241 Tocantins sequences were combined with 9289 SARS-CoV-2 genome sequences to evaluate the phylogenetic relationships of the viruses circulating in a global context. Accordingly, all genome sequences and associated metadata deposited until 31 December 2021, were downloaded from GISAID (A total of 6,665,096). The first round of subsampling was performed by using Nextstrain’s bioinformatics toolkit [23]. We selected 10,000 sequences collected from Brazil between 28 February 2020 to 31 December 2021, excluding data from Tocantins. Next, we selected 5000 sequences from South America collected between 28 February 2020 to 31 December 2021, excluding data we had already selected from Brazil. Finally, we selected 2000 sequences from Africa, Asia, China, Europe, North America, and Oceania, with the maximum date limited to 31 December 2021. For all the sequence selections, we limited the minimum-length parameter to 29,000 base pairs. These steps generated a random subsampling of 25,000 SARS-CoV-2 genomes. To reduce the dataset of genomes to allow feasible phylogenetic analysis, we then used the genome-sampler [24], which selects temporal, geographical, and diversity context samples from an available dataset containing focal sequences. Tocantins’ sequences were defined as the focal sequence collection. These steps resulted in a dataset with 9075 sequences. We also downloaded from the GISAID database 211 sequences from Tocantins state and included three sequences from December 2019 collected in Wuhan, China (EPI_ISL_402125, EPI_ISL_406798, and EPI_ISL_434534). The resulting dataset has a total of 9530 sequences.

Multiple sequence alignment was performed using MAFFT with default settings and manually inspected using AliView v1.27 [25]. The maximum likelihood (ML) phylogenetic tree was built using IQ-TREE v2.1.2 [26]. The IQ-TREE2 analysis was performed under the generalized time-reversible (GTR) model of nucleotide substitution with empirical base frequencies (+F) plus FreeRate model (+R3), as selected by the ModelFinder software [27] and 1000 replicates of ultrafast bootstrapping (−B 1000) and SH-aLRT branch test (−alrt 1000). The ML tree was inspected in TempEst v1.5.3 [28] to investigate the temporal signal through a regression analysis of root-to-tip genetic distance against sampling dates. Thirty-seven sequences were strong outliers and were removed from the analysis, leaving 9493 sequences in the final dataset. By repeating the above steps, a strong temporal signal in the final dataset was confirmed (R2 of 0.83 and correlation coefficient of 0.91; Appendix A) and a dated phylogenetic tree was inferred using TreeTime [29]. Visualizations of the time-resolved phylogenetic tree were produced in R v4.1.2 using the ggtree package [30].

Additionally, we extracted sequences of two clades that grouped a large number of Tocantins sequences from the phylogenetic tree using the caper R package [31]. Multiple sequence alignment was performed using MAFFT with default settings and manually inspected using AliView. Spatial and temporal patterns of diffusion were estimated using a Bayesian Markov Chain Monte Carlo (MCMC) approach implemented in BEAST v1.10.4 [32] with BEAGLE v3.1.2 [33] to improve computational time. The HKY + G model [34] was used to model nucleotide evolution under a strict clock model with a Continuous Time Markov Chain (CTMC) [35] and a non-parametric Bayesian Skygrid coalescent model [36]. The best-fit nucleotide substitution models for the two data sets were identified according to the Bayesian information criterion (BIC) method in jModelTest v2.1.10 [37]. Location diffusion rates were estimated using the Bayesian stochastic search variable selection (BSSVS) model [38] with a discretization scheme defined as Brazilian states. Following the Nextstrain workflow [23], the initial evolutionary clock rate was set to 8 × 10^−4^ substitutions per site per year. Two independent MCMC chains were performed with a length of 100 million generations sampling every 10,000 generations. The two independent runs were merged with Log Combiner v1.10.4 [32] and the convergence of the MCMC chain was assessed using Tracer v1.7.2 [39]. A maximum clade credibility tree (MCC) with annotated branches was then generated in TreeAnnotator v1.10.4, with 20% removed as burn-in [32]. The MCC phylogenetic trees were visualized using ggtree R package and the SpreaD3 v0.9.7.1 [40] was used to visualize the phylogeographic patterns embedded in MCC trees.

## 3. Results

Of the 241 samples from 21 December 2020 to 16 December 2021, 51.5% (*n* = 124) were from female patients, and the mean patient age was 40.3 years (interquartile range—IQR: 26 years). The mean cycle threshold (Ct) value for the RT-qPCR conducted at the Central Public Health Laboratory of the State of Tocantins was 19.3 cycles (min: 11.2, max: 26.8, median: 19.2; IQR: 3 cycles). In total, 241 samples from 46 cities from Tocantins state were used. The majority of the samples were from the capital, Palmas (*n* = 83; 34.4%), followed by Gurupi (*n* = 31; 12.9%), Araguaína (*n* = 18; 7.5%), Porto Nacional (*n* = 16; 6.6%), and Formoso do Araguaia (*n* = 8; 3.3%) (Appendix A).

### 3.1. Comparative Genome Analysis

In total, 13 Pango lineages were observed in the 241 sequenced genomes. The most frequently detected lineage was P.1.7 (78 sequences, 32.4%), followed by AY.99.2 (67 sequences, 27.8%), P.1 (56 sequences, 23.2%), AY.43 (27 sequences, 11.2%), and AY.42 (3 sequences, 1.2%). We also report here the first detection of the BA.1 lineage (VOC Omicron) in Tocantins, from a sample collected on 10 December 2021, in Gurupi municipality.

Considering the nomenclature proposed by the WHO, 136 genomes were classified as VOC Gamma and 102 genomes as VOC Delta. The genetic analysis revealed 502 different mutations comprising VOC Gamma samples, of which 52.8% (265) were non-synonymous (missense), 44.6% (224) were synonymous, 0.8% (4) were intergenic at 5′ untranslated region (UTR), 0.6% (3) were intergenic at 3′ untranslated region (UTR), and 1.2% (6) were in-frame deletions. The ORF1ab carried 62.4% (313) replacements followed by spike (S) (*n* = 73; 14.5%), ORF3a (*n* = 37; 7.4%), nucleocapsid (N) (*n* = 31; 6.2%), membrane (M) (*n* = 12; 2.4%), ORF8 (*n* = 11; 2.2%), ORF7a (*n* = 10; 2.0%), ORF7b (*n* = 6; 1.2%), ORF10 (*n* = 5; 1.0%), envelope (E) (*n* = 2; 0.4%), and ORF6 (*n* = 2; 0.4%). From the 502 different mutations, 30.7% (154) were identified in two or more sequences and 6.4% (32) were found in more than 50% of sequences (Figure 2). From these highly frequent mutations, 21 (65.6%) were non-synonymous, 9 (28.1%) were synonymous, 1 (3.1%) was intergenic at 5′ untranslated region (UTR), and 1 (3.1%) was an in-frame deletion. Among the non-synonymous mutations, 60.9% (13/21) were located in the S gene (L18F, T20N, P26S, D138Y, R190S, K417T, E484K, N501Y, D614G, H655Y, P681H, T1027I, and V1176F).

For the 102 sequences belonging the VOC Delta, 326 different mutations were found, of which 56.1% (183) were non-synonymous, 41.4% (135) were synonymous, 1.2% (4) were intergenic at 5′ untranslated region (UTR), 0.6% (2) were intergenic at 3′ untranslated region (UTR), and 0.6% (2) were in-frame deletions. Among these mutations, 60.7% (198) were found at ORF1ab followed by spike (*n* = 49; 15.0%), nucleocapsid (*n* = 26; 8.0%), ORF3a (*n* = 15; 4.6%), ORF7a (*n* = 14; 4.3%), ORF8 (*n* = 8; 2.5%), membrane (*n* = 7; 2.1%), envelope (*n* = 4; 1.2%), ORF7b (*n* = 3; 0.9%), and ORF10 (*n* = 2; 0.6%). From the total of mutations, 52.5% (171) were identified in two or more sequences. Highly frequent mutations (≥50% of genomes) were found in 34 genome positions (Figure 3). From these, 76.5% (26) mutations were non-synonymous, 11.8% (4) were synonymous, 5.9% (2) were intergenic at 5′ untranslated region (UTR), and 5.9% (2) were in-frame deletions. Remarkably, seven non-synonymous mutations were located in the S gene (T19R, G142D, L452R, T478K, D614G, P681R, and D950N).

Overall, the Tocantins comprised, until 31 December 2021, 452 SARS-CoV-2 genomes (241 sequenced in our study and 211 deposited in GISAID, until 31 December 2021). The genomic analysis with Pangolin revealed 21 SARS-CoV-2 lineages circulating in Tocantins state. The most abundant lineage was P.1, accounting for 33.4% (151 genomes) of the sequenced genomes. The second most common lineage was P.1.7 (27.0% or 122 genomes), followed by AY.99.2 (17.9% or 81 genomes), AY.43 (9.5% or 43 genomes), P.2 (3.5% or 16 genomes), and B.1.1.28 (3.1% or 14 genomes). The sequencing of SARS-CoV-2 genomes from Tocantins started in September 2020 (Figure 4). We point out here that since the emergence and spread of Gamma-related lineages (P.1 and its sub-lineages, especially P.1.7) in Brazil, they have quickly become the dominant SARS-CoV-2 lineages circulating in Tocantins state. This was a pattern clearly observed until August 2021, when the Delta lineage was introduced, and, thereafter, the sub-lineages AY.99.2, AY.43, and AY.42. During September 2021, 31 genomes were sequenced, of which 38.7% (12) belonged to the AY.99.2 lineage, followed by P.1.7 (29.0% or 9), AY.43 (9.7% or 3), and AY.42 (6.5% or 2). In October and November only, lineages associated with VOC Delta were detected among the sequenced genomes. Subsequently, in December, one genome of lineage BA.1 (VOC Omicron) was detected.

### 3.2. Phylogenetic Analysis

The time-resolved phylogenetic tree confirmed the PANGO lineages assigned, and the sequences from the Tocantins were allocated into branches close to Brazilian and South American sequences (Figure 5). We also point out that during the initial period, the predominant lineages circulating in Brazil were B.1.1.28 and B.1.1.33, followed by the rise of P.2, a descendant of the B.1.1.28 lineage, which formed a clear, evolving differentiation. Furthermore, the first sequences obtained from the Tocantins were grouped in these clades as well, as expected. Subsequently, the majority of the analyzed samples were classified as belonging to the Brazilian P.1 lineage (Gamma), which has dominated the epidemiological scenario in Brazil since its emergence (November–December 2020). Additionally, P.1.7, which originated from the P.1 lineage (spike substitution P681H), was widely spread in Tocantins from March 2021 [8]. The first confirmed case of Delta lineage in Brazil occurred on 26 April 2021. In Tocantins state, the Delta VOC was first detected in samples collected on 29 June 2021, and became dominant in September 2021. We highlight here the high spread and circulation of the Delta variant, and, in particular, of the lineage AY.99.2 in Tocantins, as well as in all Brazilian regions (Figure 5; see also Appendix A). The only BA.1 sequence identified in this study branched in a clade represented by 59 sequences, mostly represented by sequences from Australia (37.3% or 22 genomes), followed by sequences from Brazil (13.6%, or eight genomes; six from São Paulo, one from Ceará, and one from the Tocantins) (Figure 5).

We identified two subclades with high branch support, one for the P.1.7 lineage (SH-aLRT = 75.8) and one for the AY.99.2 lineage (SH-aLRT = 84.7), both grouping a large number of sequences from Tocantins, especially from our study. The P.1.7 clade was composed of 243 sequences, the majority of which were from Tocantins (49.4% or 120 genomes), followed by São Paulo (24.3% or 59 genomes) and Goiás (12.3% or 30 genomes). Furthermore, this clade was represented by 15 Brazilian states and one genome from Chile (EPI_ISL_2663617). The AY.99.2 clade was composed of 214 sequences. The majority of these were also from Tocantins (34.1% or 73 genomes). Moreover, this clade contained only Brazilian sequences, represented by 19 states and the Federal District. In order to determine the time of the most recent common ancestor (TMRCA) and the diffusion of these two subclades identified in our maximum-likelihood phylogenetic tree, we used coalescent and phylogeographic methods.

For the P.1.7 subclade, we estimated a median evolutionary rate of 4.91 × 10^−4^ (95% highest posterior density interval (HDP): 4.21 × 10^−4^ to 5.69 × 10^−4^ substitutions per site per year) and the TMRCA on 10 January 2021 (95% HPD: 27 October 2020 to 2 March 2021) (Figure 6A). Interestingly, the tree was rooted in a sample from Tocantins (EPI_ISL_2756458; PP: 1.0) and the BSSVS procedure identified well-supported rates of posterior diffusion from Tocantins to Goiás (BF: 32,149; PP: 0.99), Mato Grosso (BF: 17.54; PP: 0.55), Amapá (BF: 12.72; PP:0.47), and Pará (BF: 126.15; PP: 0.89). The tree reconstruction also showed important migrations from Tocantins to Rio Grande do Sul (BF: 18.33; PP: 0.51) and Pernambuco (BF: 10.51; PP: 0.42) (Figure 6B).

For the AY.99.2 clade, the TMRCA was dated 7 May 2021 (95% HPD: 26 March 2021 to 20 June 2021), and the median evolutionary rate was 6.20 × 10^−4^ (95% HPD: 5.37 × 10^−4^ to 7.20 × 10^−4^). The state of Rio de Janeiro was identified the most probable source of its emergence (PP = 1.0), after which multiple cluster transmission was observed across the Brazilian states, especially São Paulo, Paraiba, Federal District, and Tocantins (Figure 7A). We also found evidence of large local transmission clusters of AY.99.2 lineage in Tocantins state and secondary disseminations from Tocantins to Pará (BF: 911.27; PP: 0.98). The tree reconstruction also showed important migrations from São Paulo (BF: 336.86; PP: 0.95) and Maranhão (BF: 46.33; PP: 0.71) to Tocantins (Figure 7B).

## 4. Discussion

In this study, we report the first description of SARS-CoV-2 molecular epidemiology in Tocantins State, Brazil, from September 2020 to December 2021. We demonstrate that three main lineages dominated the SARS-CoV-2 scenario in Tocantins during this period of time: P.1 (33.4%), P.1.7 (34.9%), and AY.99.2 (17.9%). Therefore, our analysis showed that in September 2020, SARS-CoV-2 strains were mainly classified as B.1.1.28 and B.1.1.33, which is similar to observations in other Brazilian regions [2,41,42]. Nevertheless, these lineages were rapidly replaced by the variant of interest (VOI) Zeta (P.2) and, especially, by the Gamma VOC (P.1 and P.1.7), which rapidly spread and dominated the epidemiological scenarios across different regions of Brazil [8]. The P.1 lineage was first detected on 6 January 2021, by Japan’s National Institute of Infectious Disease, in four travelers who arrived in Tokyo from Amazonas, Brazil, on 2 January 2021, at airport control [3,43]. The rapid increase in the number of hospitalizations caused by this SARS-CoV-2 variant was a significant source of economic and social disruption in Brazil and led to a significant amount of pressure on the health system. AY.99.2 was a prominent ramification of VOC Delta in all of South America before the emergence of VOC Omicron [44].

The Delta VOC was first detected in the Indian state of Maharashtra in October 2020, with subsequent dissemination throughout India. The first confirmed case of Delta in Brazil occurred on 26 April 2021. A Delta lineage was first detected in Tocantins in one sample by the end of June 2021, representing 1.67% (1/60). In August 2021, it was identified in 29.8% (14/47), and in September, in more than 65% (15/23) of sequenced samples. During October and November, Delta variants became dominant in Tocantins. The observed growth in COVID-19 cases caused by Delta suggested a transition from individual to community transmission of this variant in August 2021 in Tocantins state.

The differences in viral kinetics found between the pre-Gamma, Gamma, and Delta variants in Tocantins suggest some incremental, but potentially adaptive, changes in viral dynamics. These changes are associated with the evolution of SARS-CoV-2 towards a more rapid viral spread. The emerging Brazilian Gamma variant, derived from the B.1.1.28 lineage, began spreading rapidly later in 2020 [45]. The Gamma variant spread from the north of Brazil, passing through Tocantins, towards the rest of the country. Moreover, the spread of Gamma was influenced by the demand for care for Amazon patients in other states that led to the transfer of patients to different hospitals throughout the country. Naveca et al. (2021) [46] reported that P.1+ lineages with convergent mutations evolved independently in other Brazilian states, and that the lineage designated as P.1.7 (P.1 + P681H) was the major lineage circulating outside Amazonas.

Our study indicates that the lineage AY.99.2 probably emerged in Rio de Janeiro state, and that multiple cluster transmission was observed across Brazilian states. We also point out a strong migration from Tocantins to Pará. All these observed migrations were probably influenced by the Belém–Brasília highway. In addition, the Tocantins sequences formed a separate clade, reinforcing the idea that local transmission occurred in the state. This may have been mainly influenced by the relaxation of non-pharmacological measures, such as social isolation and quarantines [46,47,48].

We recognize some limitations in the present study. First, the number of analyzed genomes corresponds to a small percentage of the number of cases. Second, the dispersal routes were influenced by multiple factors, including the geographic parameters and traffic metrics of people along the Belém–Brasília highway [49]. However, this may have been a determining factor in local transmission. Third, the dissemination of VOCs among Brazilian states is not a linear description [50,51]. The introduction of the Delta variant occurred in the first semester of 2021, during a period of restriction relaxation and incomplete vaccine schedules. Despite this, and despite the differences between states with large population concentrations (São Paulo, Rio de Janeiro), others, such as Tocantins, are influenced by the transport of agribusiness products [52,53]. In the case of the latter, the state of Tocantins stands out, as it is considered the last Brazilian agricultural frontier beyond the Amazon [11].

Our findings help to explain how the AY.99.2 lineage is transmitted so efficiently in Brazilian populations. Specifically, our results suggest that São Paulo, Federal District, Paraíba, and Tocantins states played an important role in this transmission in the Midwest, North, and Northeast regions of the country. This highlights that genomic surveillance, in addition to being useful in determining which viral variants are circulating in a given location, can also help elucidate viral trajectories. Our findings were also important to support actions to reinforce vaccination, epidemiological surveillance, and the planning of health units (intensive care units and inpatient beds), in addition to the structuring of the genomic surveillance department at the Central Public Health Laboratory of the State of Tocantins (LACEN-TO).

Despite the limitations of our study, we showed genomic surveillance to be a potential tool for monitoring the circulation of SARS-CoV-2 in Tocantins state. Although we used a relatively small number of SARS-CoV-2 sequences from Tocantins, we suggest that our analysis reflects the major variants circulating, as shown in other studies on other Brazilian states. However, levels of SARS-CoV-2 sequencing were significantly different between Brazil and other countries and across Brazilian regions and states [8]. For example, states from northern regions, such as Acre, Amapá, Roraima, and Tocantins, submitted 231, 517, 256, 452, and 890 sequences in GISAID until 31 December 2021, respectively. In comparison, states from regions in the southeast, such as São Paulo and Rio de Janeiro, submitted 45,434 and 11,081 sequences, respectively. This highlights the urgent need for implementing efficient strategies for genomic surveillance in Brazil across different regions, such as the action of the Corona-Ômica Network.Br-MCTI, allowing local public health agencies to know more about the pandemic situation in their states.

Finally, SARS-CoV-2 lineages presenting different kinetics are still emerging in the COVID-19 pandemic. We show the importance of the continuous monitoring of the epidemiology of COVID-19, as well as the specific studies of dynamics that are useful to defining public health measures. All viral surveillance information is deeply helpful in order to follow the evolution of the virus and its routes of dispersion.

## Figures and Tables

**Figure 1 viruses-14-00659-f001:**
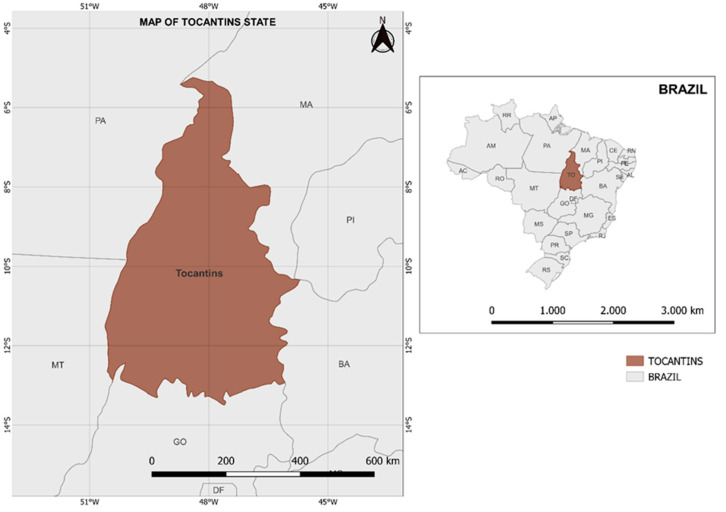
State map of Brazil with emphasis on Tocantins state. This figure was generated using QGIS v 3.20.2-Odense (Quantum Geographic Information System, QGIS Application Network Connections, Saint-Pierre, La Réunion, France).

**Figure 2 viruses-14-00659-f002:**
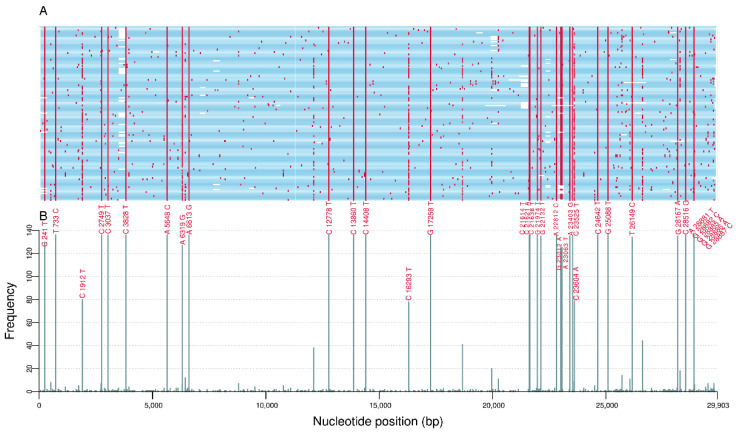
(**A**) Genomic location and (**B**) frequency of single-nucleotide polymorphisms (with respect to the reference genome NC_045512.2) among 136 sequences from VOC Gamma.

**Figure 3 viruses-14-00659-f003:**
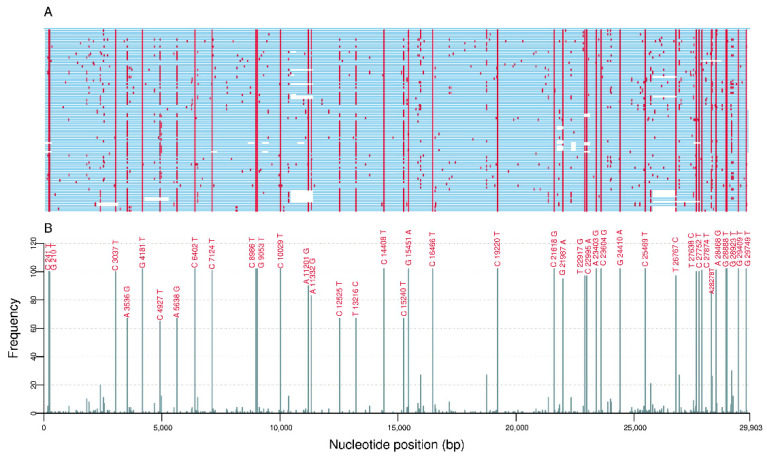
(**A**) Genomic location and (**B**) frequency of single-nucleotide polymorphisms (with respect to the reference genome NC_045512.2) among 102 sequences from VOC Delta.

**Figure 4 viruses-14-00659-f004:**
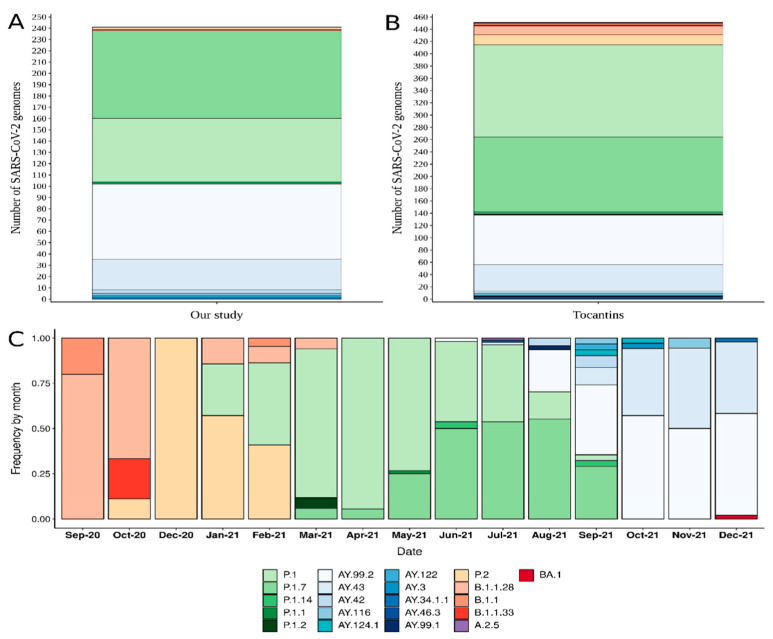
SARS-CoV-2 lineages circulating in Tocantins state. (**A**) Lineages identified in this study. (**B**) Overall lineages in the Tocantins. (**C**) Frequency of the most abundant lineages from September 2020 to December 2021.

**Figure 5 viruses-14-00659-f005:**
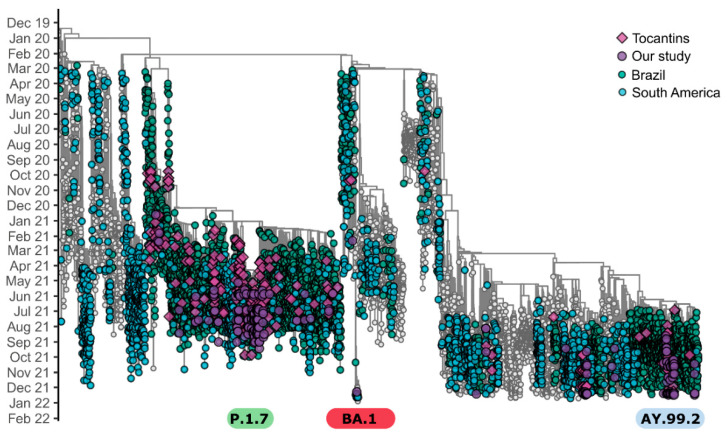
Time-resolved maximum-likelihood tree of 9493 SARS-CoV-2 genomes. Tips are colored by the most frequent Pango lineages. Diamonds represent Tocantins genomes.

**Figure 6 viruses-14-00659-f006:**
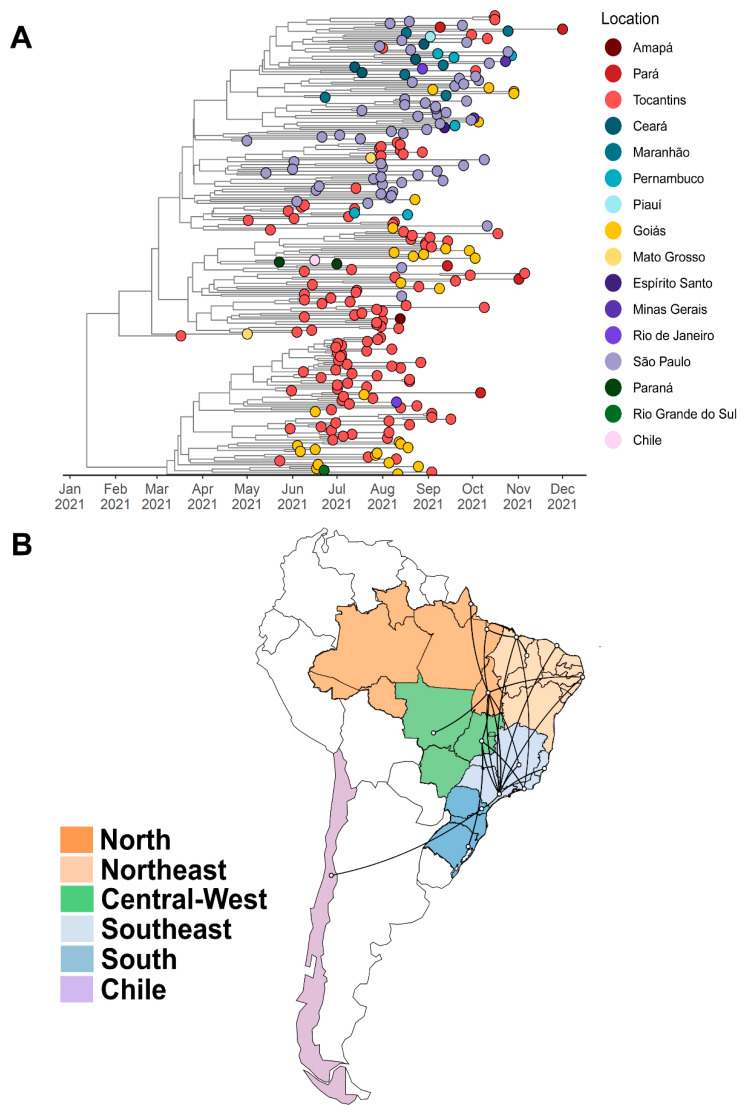
(**A**) Bayesian time-scale phylogeny of 243 P.1.7 samples circulating in Brazil from 13 March 2021, to 1 October 2021. Tips are colored based on sampling location, following the legend’s colors on the right. The molecular clock was inferred using BEAST with the HKY + G evolution model and a strict molecular clock (initial evolutionary rate was set to 8 × 10^−4^ substitutions/site/year). (**B**) Spatiotemporal diffusion of the filtered P.1.7 subclade across the Brazilian states. The figure shows well-supported transition rates of the P.1.7 lineage between Brazilian regions in discrete phylogeographic reconstructions using BSSVS procedure and Bayes Factor tests. The following states belong to each Brazilian region: North: AC, AM, AP, PA, RO, RR, and TO. Northeast: AL, BA, CE, MA, PB, PE, PI, RN, and SE. Central-West: DF, GO, MS, and MT. South: PR, RS and SC. Southeast: ES, MG, RJ, and SP. The Tocantins state is located in the North region.

**Figure 7 viruses-14-00659-f007:**
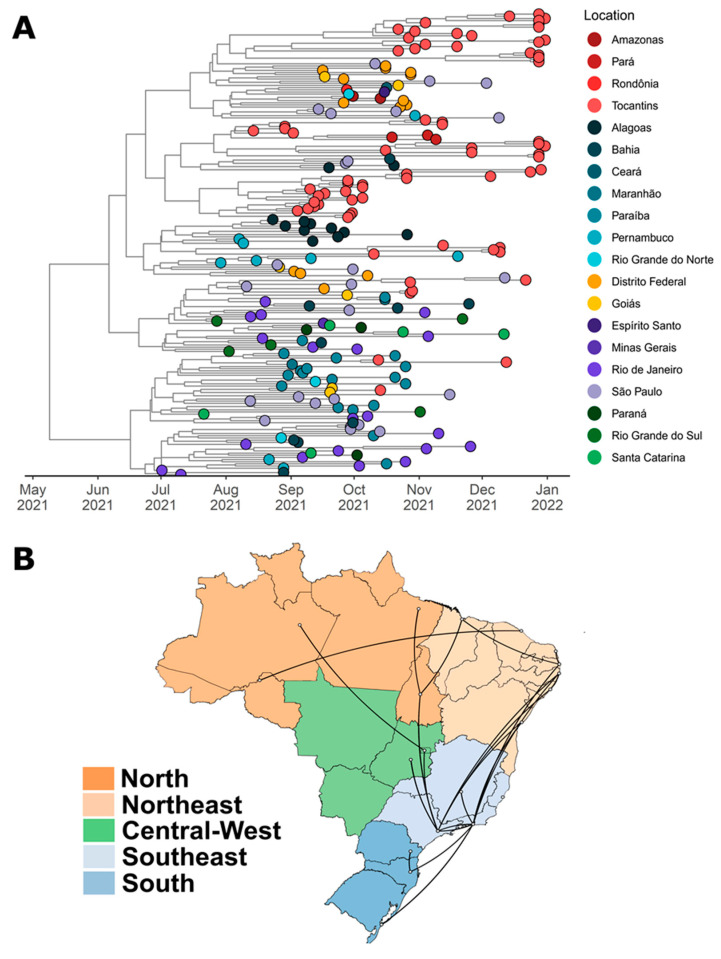
(**A**) Bayesian time-scale phylogeny of 214 AY.99.2 samples circulating in Brazil from 5 July 2021 to 16 December 2021. Tips are colored based on sampling location, following the legend’s colors on the right. The molecular clock was inferred using BEAST with the HKY + G evolution model and a strict molecular clock (initial evolutionary rate was set to 8 × 10^−4^ substitutions/site/year). (**B**) Spatiotemporal diffusion of the filtered AY.99.2 subclade across the Brazilian states. The figure shows well-supported transition rates of the AY.99.2 lineage between Brazilian regions in discrete phylogeographic reconstructions using BSSVS procedure and Bayes Factor tests.

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
