# Peer review of "Genomic Epidemiology of SARS-CoV-2 in Tocantins State and the Diffusion of P.1.7 and AY.99.2 Lineages in Brazil"

_viruses, 2022, doi:10.3390/v14040659_

Round 1

Reviewer 1 Report

The manuscript reports a very interesting work of epidemiological analysis made on genomes of SARS-CoV-2 in Tocantins State in Brazil. The methodology sounds rather solid and the results are well shown. I have only minor comments:

  • lines  44-46: this sentence seems to indicate possible causes of lack of control measures at global level. On the contrary, it is probably referring to Brazilian situation. Please clarify. Also the role of deforestation on hampering control measures is difficult to be understood.
  • line 52: when you say "alter biologically important phenotypes", can you be more specific? It is very generic.
  • line 59: together with the total number of cases in Brazil can you report the incidence for total populations ?
  • line 69: together with the total number of cases in Tocantins can you report the incidence for total populations ?
  • lines 188-189: can you provide a reference for the chosen value of the substitution rate?
  • Figure 6A: in the text the TMRA reported is December 29, 2020, but the Figure 6A starts from February 2021. In addition it is reported that the tree rooted in a sample from Sao Paulo, but this is not evident in the figure 6A, where a sample from Tocantins seems to be the close to the root.
  • lines 392-394: honestly I do not think that your study demonstrates something in relation to the link between viral load, speed of viral replication in single individuals and spread of infection. I would suggest to delete this sentence
  • line 396: please explain better what you mean with relaxation of non-pharmacological measures, possibly giving some examples.
  • line 410: your study is explaining HOW the AY.99.2 lineage spread, but I do not think that is able to explain WHY. I would suggest to delete this word from the sentence.

In conclusions, you correctly referred to some possible limitations of the study. I am wondering if a different level of sequencing in the Brazilian States may have influences the comparability of genomes obtained from the States. Please add comments for this point in the conclusions.

Reviewer 2 Report

The manuscript, submitted by de Souza and colleagues for publication to Viruses, provided an in-dept analysis on the complete genome sequences and phylogenetic analysis of 241 SARS-CoV-2 genomes detected in the Tocantins state, North Brazil, from December 2020 to December 2021. They carried out the detailed description of SARS-CoV-2 molecular epidemiology in this Brazilian state, supported by suitable statistical methods. The paper is well structured and exhaustive in every parts, while the methods is very interesting. In my opinion, it is a good example of molecular epidemiology’s application to SARS-CoV-2. It can be recommended for publication, upon addressing some minors.

In introduction, you should cite other experiences, where long-read sequencing have been used in molecular epidemiology (see 10.3390/v12091020).

At pag 4 row 167, change GRT in GTR. Moreover, at row 186 you talk about HKY+G, but how did you choose these parameters for BEAST, since you used GTR for ML? Did you calculate Bayes factor to choose the model?

At pag 3, you reported workflow for analysis on MinION output, but did you take account a further correction step on reads, because of high error rate for single read (see 10.1371/journal.pone.0257521)?

In entire paper, the word COVID-19 is written even as “Covid-19”, but I think the last one is wrong. Moreover, you should be added city and state to all companies (es. Allplex 2019- nCoV assay, Seegene Inc).
